# ANTI-REFERENCE: UNIVERSAL AND IMMEDIATE DEFENSE AGAINST REFERENCE-BASED GENERATION

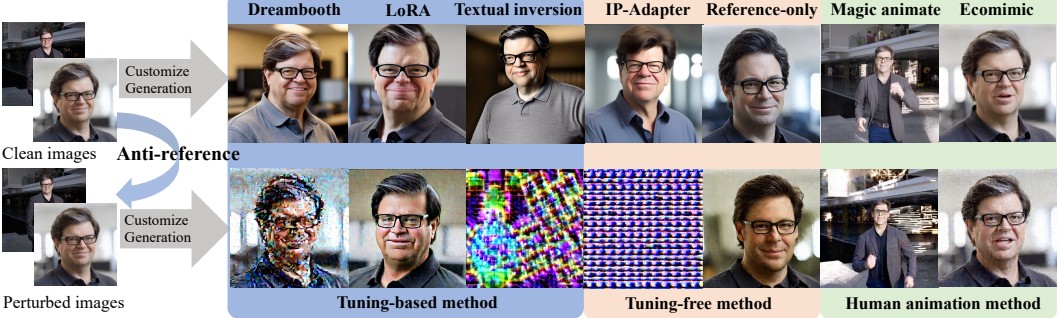

Figure 1: Malicious attackers can collect users' images as reference images and use diffusion models to achieve malicious purposes. Our system, called Anti-reference, applies imperceptible perturbations to user-uploaded images before they are published, resulting in noticeable artifacts in images or videos generated by reference-based methods and fine-tuning approaches. This makes it easy to recognize them as AI-generated, thus protecting the images.

## ABSTRACT

Diffusion models have completely transformed the field of generative models, demonstrating unparalleled capabilities in generating high-fidelity images. However, when misused, such a powerful and convenient tool could create fake news or disturbing content targeted at individual victims, causing severe negative social impacts. In this paper, we introduce Anti-Reference, a novel method that protects images from the threats posed by reference-based generation techniques by adding imperceptible adversarial noise to the images. We propose a unified loss function that enables joint attacks on fine-tuning-based customization methods, non-fine-tuning customization methods, and human-centric driving methods. Based on this loss, we train a Noise Encoder with a DiT architecture to predict the noise or directly optimize the noise using the PGD (Projected Gradient Descent) method. Our method demonstrates strong black-box transferability, being equally effective against black-box models and some commercial APIs such as Animate Anyone, and EMO. Extensive experiments validate the performance of Anti-Reference, establishing a new benchmark in image security.

## 1 INTRODUCTION

Diffusion models have completely transformed the field of generative models, demonstrating unparalleled capabilities in generating high-fidelity images. Customized generation can be divided into methods that require training, such as Dreambooth(Ruiz et al., 2023), LoRA (Hu et al., 2022), Textual Inversion (Gal et al., 2022), Custom Diffusion(Kumari et al., 2023) and those that do not, such as IP-Adapter (Ye et al., 2023a), Instant-ID (Wang et al., 2024b). Reference-based methods are widely used in customized image and video generation, especially in human-centered video generation, including portrait video creation methods(Tian et al., 2024; Chen et al., 2024; He et al., 2024; Xie et al., 2024), and human animation (Xu et al., 2024; Hu, 2024) , which have attracted significant attention due to their practical value in creating digital human avatars and enhancing film production.

However, the high convenience and efficiency of non-trainable Reference-based methods make them potentially susceptible to misuse, becoming tools for creating fake news or disturbing content targeted at individual victims, causing severe negative social impacts. Existing studies use encoder attack (Salman et al., 2023) and diffusion attack (Van Le et al., 2023; Liang et al., 2023) to protect images from the threats posed by methods requiring fine-tuning (Ruiz et al., 2023; Hu et al., 2022; Gal et al., 2022), using PGD (Madry, 2017) optimization to generate adversarial noise, but this approach requires several minutes to protect a single image, severely limiting its practical application. Moreover, these methods are largely ineffective against non-trainable Reference-based generation methods. Therefore, developing an efficient method to protect personal images from the threats of Reference-based generation has become an urgent priority.

Reference-based methods provide additional conditions through a Reference Image to enable customized generation. These methods can be divided into two types based on their implementation: one type embeds Reference features in the cross-attention layer of the denoising network using an adapter, such as IP-Adapter (Ye et al., 2023a); the other type embeds reference features in the self-attention layer of the denoising network using ReferenceNet. The approach of ReferenceNet is widely used for image customization generation (Team, 2023; Zhang et al., 2024b;c), Image2Video (Chen et al., 2023b; Zhang et al., 2023), and face animation generation (Tian et al., 2024; Chen et al., 2024; He et al., 2024; Xie et al., 2024), and body-driven tasks (Xu et al., 2024; Hu, 2024). However, due to the variety of existing Reference-based generation methods, attacking a specific method has limited practical significance, as attackers can easily switch methods to bypass protection. Therefore, the motivation of this paper is to propose a universal adversarial noise generation method to address the threats posed by mainstream Reference-based methods.

In practical image protection scenarios, protection methods need to address several challenges. Firstly, universality is a key challenge. Since Reference-based methods have many different implementations, and models trained on different datasets have different feature spaces, the same attack strategy may have very different effects on different models. Secondly, efficiency is also crucial. Existing methods like Anti-DreamBooth (Van Le et al., 2023) , which use PGD optimization, usually require hundreds of steps and significant time, severely limiting their feasibility for real-time applications. Finally, black-box transferability and robustness are also central challenges. In practical applications, the structures and parameters of proprietary APIs like EMO (Tian et al., 2024) , Animate anyone (Hu, 2024) are not accessible, so attack methods must have good black-box transferability. Additionally, the generated adversarial noise also needs to be robust enough to withstand common data augmentation operations and preprocessing steps (such as JPEG compression and Affine transformations).

To address these challenges, this paper presents Anti-Reference, the first to protect images from the threats posed by mainstream reference-based methods and tuning-based customization methods through the forward process. We propose a Noise Encoder based on the DiT (Peebles & Xie, 2023) architecture, which predicts adversarial noise of the same size as the original image and overlays it to form a protected image. To achieve a universal attack on methods requiring fine-tuning and those that do not, we designed a unified loss function, using a weighted strategy to achieve joint attack effects across multiple tasks, and by limiting the noise range and regularization loss to ensure the invisibility of the noise. To enhance the robustness of adversarial noise, we also introduced some data augmentation techniques to ensure that the adversarial noise can withstand various data enhancements and preprocessing operations. As the model structures and weights of proprietary APIs are not accessible, directly attacking these models is usually not feasible. To overcome this hurdle, we created white-box proxy models that mimic the structure and behavior of these proprietary models, and we successfully implemented attacks on these proxy models, thereby achieving black-box transferability attacks. Specifically, our adversarial samples have successfully transferred to closed-source APIs (such as Animate Anyone (Hu, 2024) and EMO (Tian et al., 2024)).

Extensive experimental results demonstrate the outstanding performance of Anti-Reference in protecting images from the threats of Reference-based generation methods and potential security risks associated with fine-tuning methods. This paper not only extends the theoretical construction of adversarial attacks but also transforms it into a deployable solution, establishing a new benchmark in the fields of privacy protection and information security.

We summarize our main contributions as follows:

- Universality and Joint Multi-Task Attacks: For the first time, we propose a universal adversarial noise generation method that employs a unified loss function to simultaneously target both mainstream Reference-based generation methods and those requiring fine-tuning.

- Efficiency Improvement: Compared to traditional PGD optimization methods, this paper introduces a DiT architecture-based Noise Encoder that enables attack execution without optimization, significantly reducing computational time and enhancing practicality for real-time applications.

- Black-Box Transferability and Robustness: We have successfully designed transferable adversarial samples that facilitate black-box attacks on proprietary commercial APIs (such as EMO, Animate Anyone) using white-box proxy models. Moreover, our method is sufficiently robust to effectively handle common data augmentation operations and complex preprocessing methods, ensuring the effectiveness of adversarial noise across various scenarios.

## 2 RELATED WORK

### 2.1 CUSTMIZED DIFFUSION MODEL.

Diffusion probability models Song et al. (2020); Ho et al. (2020) represent a class of advanced generative models that reconstruct original data from pure Gaussian noise by learning noise distributions at different levels. These models excel in handling complex data distributions and have marked significant accomplishments across various fields such as image synthesis Rombach et al. (2021); Peebles & Xie (2023), image editing Brooks et al. (2023); Hertz et al. (2022), video generation Wu et al. (2022); Hu (2024), and 3D content creation Poole et al. (2022). A prominent example is Stable Diffusion Rombach et al. (2021), which utilizes a UNet architecture to iteratively produce images, demonstrating robust text-to-image capabilities after extensive training on large text-image datasets. DreamBooth Ruiz et al. (2023), Custom diffusion Kumari et al. (2023) and Textual Inversion Gal et al. (2022), adopt transfer learning to text-to-image diffusion models via either fine-tuning all the parameters, partial parameters , or introducing and optimizing a word vector for the new concept. LoRA (Low-Rank Adaptation) Hu et al. (2022) is a popular and lightweight training technique that significantly reduces the number of trainable parameters and is widely used for personalized or task-specific image generation.

### 2.2 REFERENCE-BASED GENERATION

In addition to the aforementioned fine-tuning methods, finetuning-free concept learning methods can capture concepts from a single image and are widely used for tasks such as customized generation (Ye et al., 2023a; Zhang et al., 2024a), identity consistency maintenance (Wang et al., 2024b; Li et al., 2024), face-driven Tian et al. (2024); Chen et al. (2024); Xie et al. (2024), and body-driven tasks Xu et al. (2024); Hu (2024). These methods can be roughly categorized into the Adapter approach and the ReferenceNet approach based on how the reference image features are utilized. In the Adapter approach, the reference image is first processed by a pre-trained image feature extractor, typically CLIP (Radford et al., 2021) image encoder or ArcFace Deng et al. (2019), and then an adapter structure generates visual tokens applied to the cross-attention layers of the U-Net. The ReferenceNet approach emphasizes the effectiveness of integrating reference image features into the self-attention layers of LDM U-Nets, enabling customized generation while preserving appearance context. Image-to-video technology Chen et al. (2023b); Zhang et al. (2023) uses ReferenceNet to maintain consistency between the generated results and the reference image. Magic Animate Xu et al. (2024) and Animate Anyone Hu (2024) combine ReferenceNet with pose control and temporal modules to achieve body-driven generation. EMO Tian et al. (2024), Ecomimic Chen et al. (2024), and X-Portrait Xie et al. (2024), among other talking-face methods, maintain identity consistency using ReferenceNet, generating fake videos from just a single photo. The misuse of Reference-based Generation methods can have severe consequences, making it urgent to protect images from the threats posed by such methods.

## 2.3 PROTECTIVE PERTURBATION AGAINST DIFFUSION.

Protective Perturbation against Stable Diffusion. To protect personal images such as faces and artwork from potential infringement when used for fine-tuning Stable Diffusion, recent research aims to disrupt the fine-tuning process by adding imperceptible protective noise to these images. Several methods have been developed to achieve this goal: Glaze (Shan et al., 2023) focuses on preventing artists' work from being used for specific style mimicry in Stable Diffusion. It optimizes the distance between the original image and the target image at the feature level, causing Stable Diffusion to learn the wrong artistic style. AdvDM (Liang et al., 2023) proposes a direct adversarial attack on Stable Diffusion by maximizing the Mean Squared Error loss during the optimization process. This approach uses adversarial noise to protect personal images. Anti-DreamBooth (Van Le et al., 2023) incorporates the DreamBooth fine-tuning process of Stable Diffusion into its consideration. It designs a bi-level min-max optimization process to generate protective perturbations. Additionally, other research efforts (Wang et al., 2024a; Ye et al., 2023b; Zheng et al., 2023) have explored generating protective noise for images using similar adversarial perturbation methods. The aforementioned works use PGD optimization to protect images from threats posed by fine-tuning methods such as Dreambooth (Ruiz et al., 2023), LoRA (Hu et al., 2022), and textual inversion (Gal et al., 2022). However, these methods fail to provide protection against reference-based generation. Therefore, it is urgent to explore how to protect personal images from reference-based generation, and this paper fills that gap.

| Generation Task Categories | Examples of Methods |
|---|---|
| Adapter-based image generation | IP-Adapter Ye et al. (2023a), InstantID Wang et al. (2024b), Anydoor Chen et al. (2023a), SSR-Encoder Zhang et al. (2024a) |
| Reference-based image generation | Reference-only Team (2023), Stable-Makeup Zhang et al. (2024b), StableHair Zhang et al. (2024c) |
| Reference-based video generation | Animate anyone Hu (2024), Magicanimate Xu et al. (2024), EMO Tian et al. (2024), EchoMimic Chen et al. (2024), X-Potrait Xie et al. (2024), Echo-PMO Tian et al. (2024) |

Table 1: Overview of Generative Methods by Category

## 3 PROBLEM DEFINITION

Given the practical implications of image infringement based on Stable Diffusion, it is essential to define the threat model in real-world scenarios. We consider two participants involved in fine-tuning Stable Diffusion using images: the "image protector" Alice and the "image exploiter" Bob. Bob illicitly uses reference-based methods to exploit others' photos for customized content, while Alice, wishing to safeguard her images on social media, adds adversarial noise to disrupt Bob's methods, aiming to induce severe artifacts in the generated content. Specifically, we explain the workflow of the two parties as follows:

**Image Protector Alice:** The Image Protector aims to provide protection for images to prevent exploitation by Stable Diffusion. In this context, the chosen protection method involves adding imperceptible protective perturbations to the images, with the goal of offering protection while minimizing alterations to the original image. In real-world scenarios, the Image Protector often faces challenges, such as not knowing the methods and forms the Image Exploiter will use to fine-tune Stable Diffusion with the protected images. Additionally, they cannot protect images that have been publicly disclosed in the past.

**Image Exploiter Bob:** The Image Exploiter aims to fine-tune Stable Diffusion using images collected from the internet to generate high-quality images with specific concepts, including faces, objects, and artistic styles. To realistically assess the effectiveness of protective perturbations, we consider that the Image Exploiter may have the following possibilities during image collection and fine-tuning: (1) The Image Exploiter can choose any fine-tuning method, including but not limited to direct fine-tuning, LoRA, Textual Inversion, DreamBooth, and Custom Diffusion, among other mainstream fine-tuning methods. This requires the Image Protector to ensure that the protected images remain effective against any fine-tuning method. (2) The protected images may undergo natural transformations during the dissemination process, including but not limited to cropping, compression, and blurring. This necessitates the Image Protector to consider the robustness of protective perturbations when exposed to these natural disturbances. (3) Image pre-processing: The Image Ex-

ploiter may employ purification methods to remove the protective perturbations from the collected images after acquisition.

The goal of this work is to add imperceptible adversarial noise to images, formalized as $I' = I + \text{noise}$, where $I$ and $I'$ represent the original and protected images, respectively. These images serve as inputs to customization methods, and the outputs $\text{Gen}(I)$ and $\text{Gen}(I')$ are compared. If $\text{Gen}(I')$ exhibits significant distortion, the protection is considered successful. We achieve this by solving the following optimization problem:

$$\max_{x_{adv} \in M} d(\text{Gen}(I), \text{Gen}(I_{adv})) \text{ subject to } d'(I, I_{adv}) \leq \delta,$$

where $M$ indicates the natural image manifold, $d$ and $d'$ denote image distance functions, and $\delta$ represents the fidelity budget. Through this optimization process, we aim to effectively safeguard images from unauthorized editing and translation while maintaining their fidelity.

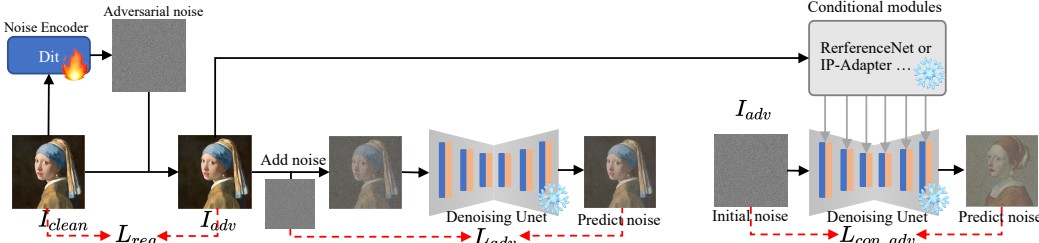

Figure 2: Illustration of Anti-reference. We propose a loss function to protect images from the threats of customized generation methods, and we use this loss to train a noise encoder to predict adversarial noise.

## 4 METHOD

In Section 4.1, we first introduce the overall method. In Section 4.2, we introduce the Noise Encoder in detail. In Section 4.3, we introduce the Loss function we use. In Section 4.4, we describe how to implement PGD joint optimization. In Section 4.5, we explain how to create white-box proxies for black-box models to facilitate attacks.

### 4.1 OVERALL METHOD

This section introduces the overall framework of the Anti-Reference method, as shown in Figure 2. Our method consists of several key components: the Noise Encoder, a set of conditional modules, the Denoising Unet, and a differentiable data augmentation module. The Noise Encoder adds adversarial noise to the image, forming the protected image $I_{adv}$. The set of Reference Modules is a group of conditional control modules that serve as the target models for the attack.

To protect images from the threats posed by tuning-free customization generation methods and driving methods, we selected the pre-trained ReferenceNet from Magic Animate and Ecomimic, as well as the Stable Diffusion Unet, as the target models for attacking the ReferenceNet route. Additionally, we chose the IP-Adapter as the target model for the Adapter route. The Denoising Unet utilizes the pre-trained Stable Diffusion 1.5 Unet, as it is the most commonly used base model for various customization generation methods. The protected image $I_{adv}$ is fed into two components: the set of conditional modules and the Denoising Unet, where losses are calculated separately. To enhance the robustness of the adversarial noise against real-world scenarios, we propose a differentiable data augmentation module, which applies common data augmentations to $I_{adv}$.

## 4.2 Noise Encoder

Our objective is to input an image that needs protection, and the Noise Encoder generates adversarial noise of the same size as the image through a forward process. In designing the structure of the noise encoder, we initially tried the hidden watermark embedding method proposed by Hidden and the U-Net structure from stable diffusion, but neither yielded satisfactory results. Ultimately, we proposed the Noise Encoder based on the DiT structure. DiT, a new diffusion model based on transformers, is referred to as Diffusion Transformers and follows the best practices of Vision Transformers (ViTs). ViTs have demonstrated superior scalability in visual recognition tasks compared to traditional convolutional networks like ResNet. Specifically, the DiT model processes input images by patchifying them into smaller blocks, which are then fed as input sequences into transformers. After being processed by multiple transformer blocks, the noise is gradually removed, resulting in high-quality image generation.

In our Noise Encoder design, there are several key differences from the DiT design: 1. Our Noise Encoder generates adversarial noise in a single inference rather than iterative denoising. 2. We removed the additional conditional information related to the diffusion process from DiT, such as noise timesteps t, class labels c, and natural language. 3. DiT performs denoising in the latent space of the VAE, whereas our method predicts adversarial noise in the pixel domain, with differences in input channels and resolution. Considering that the customization methods targeted by our attacks typically use SD1.5 as the base model, with the input image resolution for ReferenceNet and the denoising U-Net set to 512×512, we also set the training resolution of DiT to 512×512 to ensure compatibility with these customized generation tasks and to enhance the effectiveness of the attack.

To ensure the adversarial noise retains its effectiveness under various image processing conditions, we apply a series of data augmentations before utilizing the noised image $I'$. These augmentations include differentiable cropping, resizing, JPEG compression, and color adjustments, all of which help maintain the robustness of the adversarial noise against common image transformations.

## 4.3 Loss Function

**Diffusion adv loss** In adversarial attacks, our goal is to maximize the noise prediction loss of the diffusion model, rather than minimize it. This means that we aim for the noise predicted by the model, $\epsilon_\theta$, to have the largest possible error compared to the actual noise $\epsilon$, thereby disrupting the model's denoising capability. The specific loss function can be defined as:

$$L_{\text{adv}} = -\mathbb{E}_{x_0, \epsilon \sim \mathcal{N}(0,1), t} \left[ \|\epsilon - \epsilon_\theta(x_t, t)\|^2 \right]$$

Where $x_0$ is the original data, $\epsilon$ is noise sampled from a standard normal distribution, $t$ is the time step representing the noise level, $x_t = \sqrt{\bar{\alpha}_t} x_0 + \sqrt{1 - \bar{\alpha}_t} \epsilon$ is the noisy image at time step $t$, $\epsilon_\theta(x_t, t)$ is the noise predicted by the model. This loss function is as same as diffusion training loss, but the objective is completely opposite.

**Conditional Adversarial Loss** Conditional Adversarial Loss aims to attack reference-based customization generation methods and driving techniques. Specifically, we calculate the adversarial noise prediction loss when adversarial noise images are used as inputs for ReferenceNet or IP-adapter. This loss measures the deviation of the noise predicted by the denoising Unet from the actual noise, under specific conditional features provided by either ReferenceNet or the IP-adapter. The conditional adversarial loss is formulated as follows:

$$L_{\text{con\_adv}} = -\mathbb{E}_{x_0, \epsilon \sim \mathcal{N}(0,1), t, c} \left[ \|\epsilon - \epsilon_\theta(x_t, t, c)\|^2 \right]$$

$c$ represents the features extracted from $I_{adv}$ using ReferenceNet or IP-adapter. These features disrupt the denoising process by interacting with the cross-attention or self-attention layers of the Denoising Unet.

**Image Regularization Loss** To make the adversarial noise less perceptible, we calculate the Mean Squared Error (MSE) of the images before and after noise addition as the regularization loss.

$$L_{\text{reg}} = \text{MSE}(I, I_{adv})$$

**Total Loss** For joint attacks, a weighted loss formulation is employed to ensure a balanced attack performance across various tasks by balancing the impact across all contributions. The total loss, incorporating adversarial, conditional adversarial, and regularization losses, is defined as follows:

$$L_{\text{total}} = w_{\text{adv}} \cdot L_{\text{adv}} + \sum_i w_{\text{con},i} \cdot L_{\text{con\_adv},i} + w_{\text{reg}} \cdot L_{\text{reg}}$$

Here, $w_{\text{con},i} \cdot L_{\text{con\_adv},i}$ represents the weighted sum of conditional adversarial losses from different conditional modules. Each module $i$ targets different conditional control tasks, and $w_{\text{con},i}$ is the specific weight assigned to the conditional adversarial loss for module $i$. This paper conducts joint training across four conditional modules: IP-Adapter, Reference-only, Magic Animate, and Ecomimic's ReferenceNet. This approach allows for tailored defenses against a range of adversarial manipulations facilitated by different attack modules, ensuring that the influence of each module is properly scaled according to its significance and effectiveness in the overall defense strategy.

### 4.4 PGD Joint Optimization

We introduce our Anti-Reference (PGD) method, where adversarial noise is optimized directly using PGD (Projected Gradient Descent). PGD iteratively perturbs the input image $I$ within a predefined bound, ensuring the noise remains imperceptible while maximizing its impact on the model's predictions. Unlike the Noise Encoder, which generates noise in a single pass, PGD updates the noise iteratively by calculating the gradient of the loss function with respect to the image. At each iteration, the adversarial noise is updated as:

$$I_{adv}^{(k+1)} = \Pi_{I+\epsilon} \left( I_{adv}^{(k)} + \alpha \cdot \text{sign} \left( \nabla_{I_{adv}^{(k)}} L_{\text{total}} \right) \right)$$

Here, $I_{adv}^{(k)}$ is the adversarial image at iteration $k$, $\alpha$ is the step size, and $\epsilon$ defines the perturbation bound. The projection $\Pi_{I+\epsilon}$ ensures the noise stays within the allowed limits.

By optimizing both $L_{\text{adv}}$ and $L_{\text{con\_adv}}$, PGD effectively disrupts both the diffusion process and conditional adversarial predictions. Our experiments show that PGD provides strong protection across various reference-based customization methods, with gradually increasing noise impact while preserving image quality. Although Noise Encoder generates noise faster, PGD's iterative process delivers stronger protection across tasks, albeit with higher computational costs, making it suitable for scenarios requiring maximum protection.

### 4.5 Black-Box Transfer

This section introduces proxy-based black-box attacks, a method that generates adversarial samples using a white-box model with a structure similar to the target black-box model or a closely related latent space. By training DiT to generate adversarial samples on the white-box model, these samples also achieve high attack success rates on the black-box model. The success of this approach relies on two key factors: 1) structural similarity between the white-box and black-box models, and 2) shared similarity in their latent spaces. For instance, both Animate Anyone and Magic Animate are based on Stable Diffusion 1.5 and share the same ReferenceNet architecture, with similar datasets used for fine-tuning, resulting in similar latent spaces. Additionally, we successfully attacked the EMO, Animate anyone and other apps or APIs, as demonstrated in the experiments.

## 5 Experiment

### 5.1 Setup

**Training data** This paper aims to achieve general image protection, and therefore, we use 600K natural image-text pairs from the Laion dataset as the training set. To enhance the protection effectiveness for talking face and body-driven tasks, we also include the Celeb-A dataset (200K) and the TikTok dataset (30K) into the training data.

**Experimental details** We used 4 A100 GPUs to train on 830,000 image-text pairs for 4 epochs with a batch size of 8, employing a learning rate decay strategy with an initial value of $10^{-3}$. We utilized a pre-trained DiT-S/8 model as the pre-trained model for the Noise Encoder.

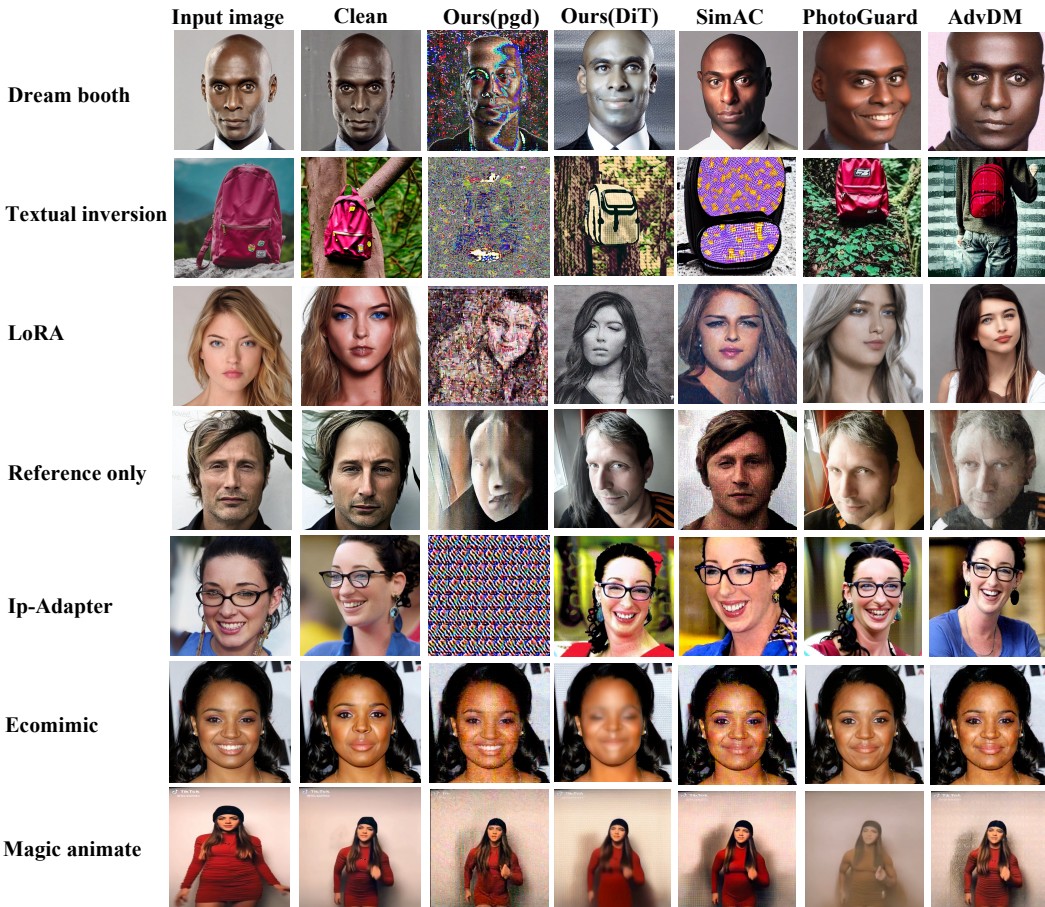

Figure 3: Results of different image protection methods in safeguarding images from the threats of customized generation tasks.

**Baselines and Evaluation benchmarks** We use PhotoGuard (Salman et al., 2023), AdvDM (Liang et al., 2023), and SimAC (Wang et al., 2024a) as baselines, with SimAC being an improved version of the classic Anti-DreamBooth (Van Le et al., 2023). We systematically evaluate the protection effectiveness of our method and the baseline methods across seven customization generation tasks, including three fine-tuning-based methods: DreamBooth, LoRA, and Textual Inversion; two tuning-free methods: IP-Adapter and reference-only; and two tasks involving human figure animation: Magic Animate and Ecomimic.

**Evaluation benchmarks** In constructing the evaluation dataset, we follow previous works. For subject-driven generation, we select 10 subject categories from the DreamBooth dataset [18], with 3 to 5 images per category. For face-driven tasks, we use 10 identities from the CelebA-HQ dataset. For each subject or individual, we generate a total of 200 images using 10 different prompts for quantitative evaluation. For face-driven and body animation tasks, we generate 200 images using CelebA-HQ and TikTok data, respectively, for quantitative comparison.

**Evaluation metrics** In our evaluation of person-centric image generation quality, we utilized the FDR (Face Detection Rate) and ISM (Identity Score Matching) metrics (Van Le et al., 2023) to assess protection effectiveness, where lower FDR indicates a significant reduction in generative effect, making it difficult for face detection technologies to detect, and lower ISM scores indicate more effective disruption of individual identity in the generated images. Additionally, we measured general

image quality using Aesthetics Score (AI, 2023) and CLIP-IQA (CLIP Image Quality Assessment) (Wang et al., 2023), which evaluate the naturalness and perceptual quality of images. These metrics were applied across all frames for tasks involving human body and face-driven content. Lower values in these metrics indicate better image protection effectiveness.

## 5.2 QUANTITATIVE EVALUATION

In this section, we present the quantitative evaluation results and time cost for our method and baselines across seven customized generation methods. For all baseline methods, we use their default code and settings to learn adversarial noise. The results of our two methods used for calculating quantitative metrics are all obtained through joint optimization while results of other baselines are optimized independently on each generation method.

**Effectiveness.** From Figures 3 and 4, it is evident that our two methods exhibit more comprehensive and thorough attack effects compared to the baseline. Our PGD method effectively protects images from the threats of 7 customized generation methods, and our DiT method also demonstrates effectiveness across all tasks. Specifically, PhotoGuard performed best in the FDR metric, closely followed by our PGD and DiT methods. In the ISM metric, both our PGD and DiT methods achieved leading results. Regarding the Aesthetic Score, our PGD method showed a definitive advantage, followed closely by the SimAC method and our DiT method. On the CLIP-IQA metric, our two methods demonstrated superiority in protecting against training-based methods.

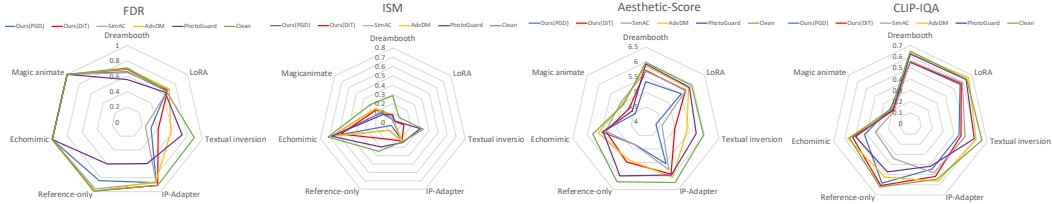

Figure 4: Comprehensive Evaluation of 7 Customized Generation Methods Across 4 Metrics.

**Time Cost.** Figure 6 shows a comparison of the time required to protect a single image using our method versus the baseline methods. Our method takes only one thousandth of the time required by the baseline methods. This improvement in efficiency marks a crucial advancement from academic research to practical application, laying the foundation for real-world implementation in AI security.

| Method | GPU Time (s) | CPU Time (s) |
|---|---|---|
| Ours(PGD) | 846 | - |
| Ours(DiT) | 0.21 | 1.05 |
| AdvDM | 212 | - |
| PhotoGuard | 66 | - |
| SimAC | 51 | - |

Figure 5: Time Cost Comparison.

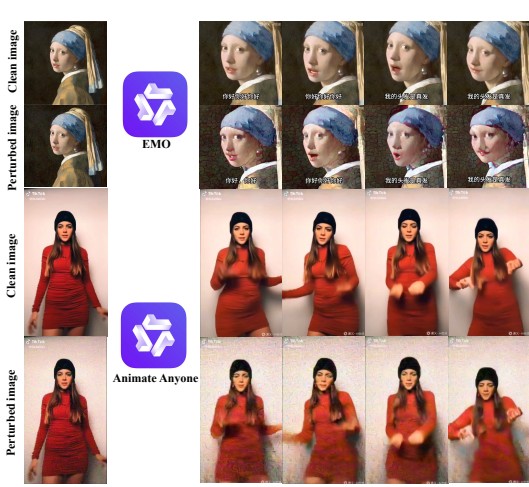

Figure 6: Black-box attack results.

## 5.3 QUALITATIVE EVALUATION

**Black-Box Performance** In this section, we demonstrate the black-box transferability of our method. We tested the closed-source face-driven method EMO (Tian et al., 2024) and body-driven method Animate anyone (Hu, 2024) on the Tongyi app. In all of the aforementioned software and commercial APIs, we were unable to access the model parameters or any internal information. From the results, our method demonstrates excellent black-box transferability, with noticeable artifacts appearing in the customized generation outputs. Additionally, these APIs require cropping of input images, and our method demonstrates robustness to cropping.

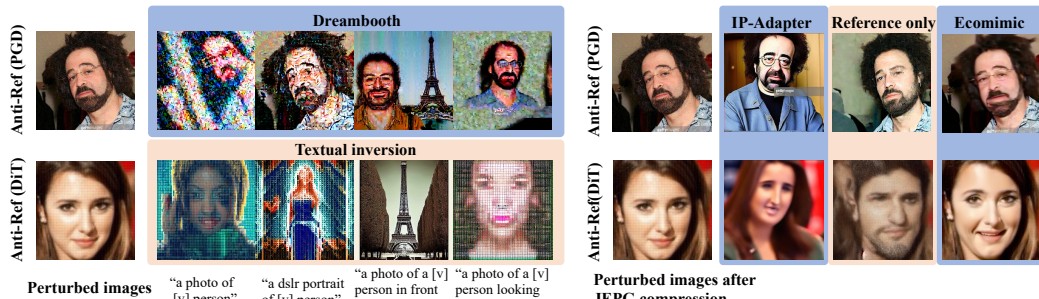

Figure 7: Qualitative Evaluation of Method Robustness: Performance Under Prompt Mismatch and JPEG Compression.

**Robustness Test** In this section, we qualitatively evaluate the robustness of our method, including its performance under prompt mismatch and JPEG compression.

- **Prompt Mismatch.** When Bob uses the Stable Diffusion model to customize concepts, the prompts he uses might differ from the assumptions Alice made when adding noise. Current PGD-based optimization methods (Van Le et al., 2023),which use "a photo of sks person" during perturbation learning, exhibit performance degradation when faced with different prompts during inference. As shown in Figure 7, our method, which trains DiT on a large-scale image-text dataset, is inherently robust to different prompts.

- **JPEG Compression** JPEG compression is the most common operation in image transmission, and Figure 7 demonstrates the robustness of our method against the JPEG compression.

## 6 LIMITATIONS AND FUTURE WORK

There are many reference-based customized generation methods, and our model trained on SD1.5 cannot handle architectures like SD-XL, SD3, or image generation methods based on autoregressive models. In the future, joint training across different architectures may be a solution. The DiT scheme proposed in this paper still has some gaps in effectiveness compared to the results optimized with PGD on a single image. In addition, we will explore ways to make the adversarial noise less perceptible in the future.

## 7 CONCLUSION

This paper introduces Anti-Reference, a novel and effective method for protecting images from the threats posed by mainstream Reference-based generation methods and fine-tuning-based methods. Utilizing a Noise Encoder based on the DiT architecture and a unified loss function, our approach offers universal and efficient protection against various adversarial attacks. Additionally, the introduction of data augmentation techniques and black-box transfer capabilities through white-box proxy models ensures robust and scalable defenses. Extensive experiments validate the effectiveness of Anti-Reference in protecting images from unauthorized customized generation, setting a new standard in the fields of privacy protection and information security.

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

# A APPENDIX

## A.1 CRITICAL OVERSIGHT

It is worth noting that when training Dreambooth with adversarial images, we did not fine-tune the CLIP text encoder, which aligns with the common practice in the community. We found that the good protection performance of Anti-Dreambooth and SimAC is based on the incorrect assumption that Bob will fine-tune the CLIP text encoder. As shown in Figure 7, when the attacker Bob does not fine-tune the CLIP text encoder during Dreambooth training, both of these image protection methods show a significant drop in performance, regardless of whether the CLIP text encoder was fine-tuned during the noise learning process. Our method does not suffer from this issue.

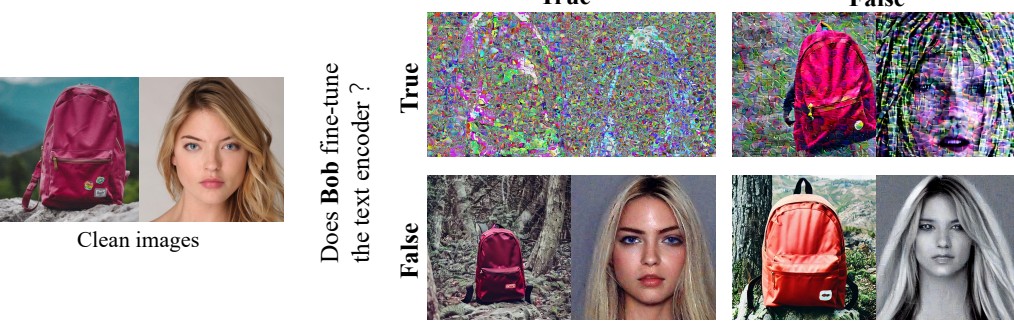

Figure 8: We have identified a critical oversight in the current SimAC method; when Bob does not train the Text Encoder while training Dreambooth, the protection effectiveness of the images is significantly compromised.

## A.2 DETAIL OF EVALUATION METRICS

For evaluating the quality of person-centric image generation, we used widely adopted metrics FDR and ISM (Van Le et al., 2023) to quantify the generation quality, where higher FDFR and lower ISM represent better protection effectiveness. Additionally, we employed two general image quality assessment metrics, Aesthetic Score (AI, 2023) and CLIP-IQA (Wang et al., 2023). For human body and face-driven tasks, we calculated quantitative metrics across all frames.

- **FDR** (Face Detection Rate): Measures the effectiveness of face detection by calculating the percentage of images in which a face detector, such as RetinaFace, successfully detects a face. A lower success rate indicates better image protection effectiveness.
- **ISM** (Identity Score Matching): Measures the cosine similarity between the features of the generated face and the original face to evaluate how well the generated image maintains the identity of the subject.
- **Aesthetic Score**: An aesthetic assessment metric that utilizes a linear estimator built on top of CLIP to predict the aesthetic quality of images.
- **CLIP-IQA** (CLIP Image Quality Assessment): Uses CLIP (Contrastive Language-Image Pretraining) to evaluate the perceptual quality of images by assessing how well the visual features of the image align with text descriptions.

## A.3 INVISIBILITY OF ADVERSARIAL NOISE.

Table 4 provides a comparison of the invisibility of adversarial noise introduced by our method versus baseline methods, measured using Mean Squared Error (MSE), Peak Signal-to-Noise Ratio (PSNR), and Structural Similarity Index (SSIM). Lower MSE and higher PSNR and SSIM values

Table 2: FDR($\downarrow$) .

| Method | Ours(PGD) | Ours(DiT) | SimAC | AdvDM | PhotoGuard | Clean |
|---|---|---|---|---|---|---|
| Dreambooth | 0.658 | 0.690 | 0.646 | 0.698 | 0.556 | 0.708 |
| LoRA | 0.656 | 0.672 | 0.688 | 0.696 | 0.628 | 0.650 |
| Textual Inversion | 0.318 | 0.412 | 0.242 | 0.582 | 0.732 | 0.894 |
| IP-Adapter | 0.870 | 0.910 | 0.870 | 0.855 | 0.595 | 0.910 |
| Reference-only | 0.840 | 0.990 | 0.960 | 1.000 | 0.600 | 0.990 |
| Echomimic | 1.000 | 1.000 | 1.000 | 1.000 | 1.000 | 1.000 |
| Magic Animate | 1.000 | 1.000 | 1.000 | 1.000 | 1.000 | 1.000 |

Table 3: ISM($\downarrow$).

| Method | Ours(PGD) | Ours(DiT) | SimAC | AdvDM | PhotoGuard | Clean |
|---|---|---|---|---|---|---|
| Dreambooth | 0.029 | 0.078 | 0.051 | 0.077 | 0.081 | 0.287 |
| LoRA | 0.005 | 0.017 | 0.008 | 0.015 | 0.022 | 0.085 |
| Textual Inversion | 0.011 | 0.123 | 0.018 | 0.018 | 0.304 | 0.336 |
| IP-Adapter | 0.197 | 0.226 | 0.225 | 0.225 | 0.242 | 0.233 |
| Reference-only | 0.038 | 0.198 | 0.096 | 0.096 | 0.295 | 0.348 |
| Echomimic | 0.655 | 0.574 | 0.673 | 0.668 | 0.677 | 0.715 |
| Magic Animate | 0.163 | 0.221 | 0.236 | 0.236 | 0.134 | 0.308 |

Table 4: Aesthetic Score($\downarrow$).

| Method | Ours(PGD) | Ours(DiT) | SimAC | AdvDM | PhotoGuard | Clean |
|---|---|---|---|---|---|---|
| Dreambooth | 5.345 | 5.716 | 5.687 | 5.874 | 5.935 | 5.985 |
| LoRA | 5.511 | 5.694 | 5.719 | 5.823 | 5.856 | 5.951 |
| Textual Inversion | 4.344 | 4.988 | 4.552 | 5.4 | 5.723 | 5.971 |
| IP-Adapter | 5.548 | 5.93 | 5.771 | 6.05 | 5.961 | 6.241 |
| Reference-only | 4.836 | 5.48 | 4.847 | 5.384 | 5.996 | 6.216 |
| Echomimic | 5.506 | 5.37 | 5.377 | 5.631 | 5.461 | 5.817 |
| Magic Animate | 4.451 | 4.716 | 5.057 | 4.988 | 4.582 | 4.951 |

Table 5: CLIP-IQA ($\downarrow$).

| Method | Ours(PGD) | Ours(DiT) | SimAC | AdvDM | PhotoGuard | Clean |
|---|---|---|---|---|---|---|
| Dreambooth | 0.550 | 0.552 | 0.561 | 0.631 | 0.623 | 0.648 |
| LoRA | 0.566 | 0.579 | 0.591 | 0.662 | 0.634 | 0.642 |
| Textual Inversion | 0.444 | 0.462 | 0.500 | 0.599 | 0.583 | 0.653 |
| IP-Adapter | 0.445 | 0.517 | 0.483 | 0.566 | 0.416 | 0.545 |
| Reference-only | 0.584 | 0.608 | 0.341 | 0.523 | 0.473 | 0.622 |
| Echomimic | 0.419 | 0.527 | 0.319 | 0.573 | 0.500 | 0.556 |
| Magic Animate | 0.225 | 0.202 | 0.184 | 0.191 | 0.196 | 0.217 |

indicate better invisibility of the noise. Our method achieves a lower MSE and higher PSNR and SSIM values compared to the baseline methods, demonstrating superior performance in maintaining the visual quality of the image while effectively applying adversarial noise.

Table 6 provides a comparison of the invisibility of adversarial noise between our method and baseline methods, using metrics including Peak Signal-to-Noise Ratio (PSNR), and Structural Similarity Index (SSIM). Higher PSNR and SSIM values indicate better noise invisibility. AdvDM exhibits the best noise invisibility, while our method achieves comparable invisibility levels to the other two baseline methods.

Table 6: Comparison of Adversarial Noise Invisibility.

| Method | PSNR (dB) ($\uparrow$) | SSIM ($\uparrow$) |
|---|---|---|
| Ours(PGD) | 30.39 | 0.762 |
| Ours(DiT) | 29.00 | 0.713 |
| AdvDM | 38.04 | 0.939 |
| PhotoGuard | 32.25 | 0.822 |
| SimAC | 32.17 | 0.811 |

## A.4 ADDITIONAL RESULTS OF OUR METHODS

Figure 9, 10 and 11 show additional attacking results of our methods on seven pipelines. Note that results for both PGD and DiT are obtained through joint optimization on four conditional modules as stated in Section 4.3.

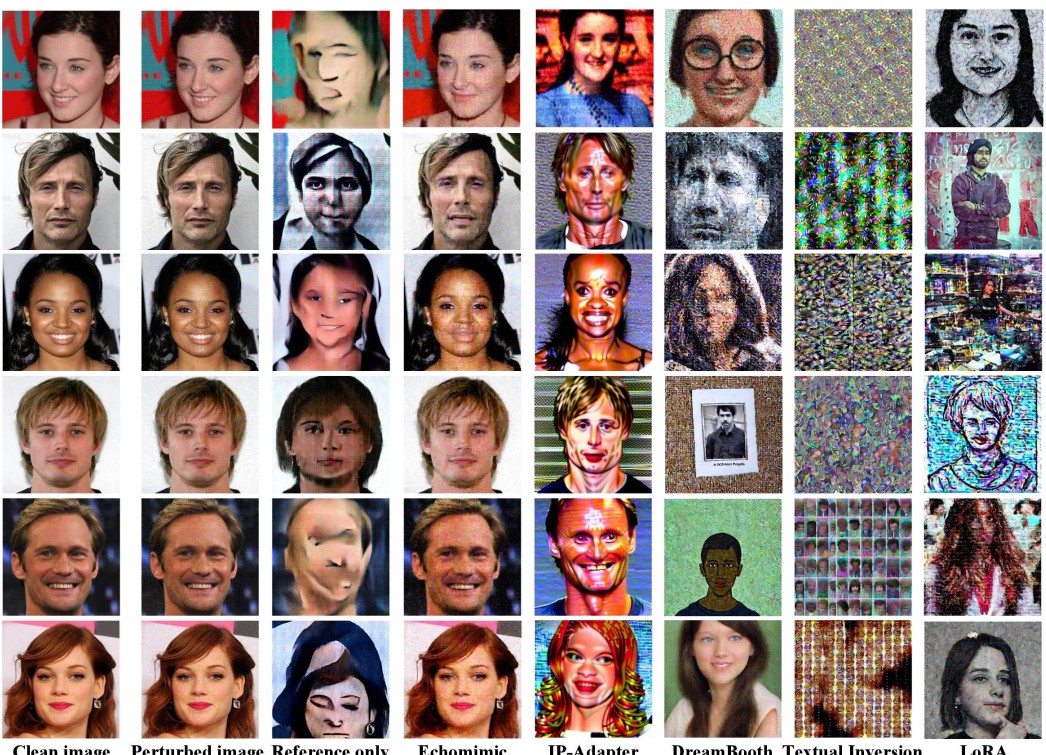

Clean image    Perturbed image    Reference only    Echomimic    IP-Adapter    DreamBooth    Textual Inversion    LoRA

Figure 9: Additional results of our method (PGD).

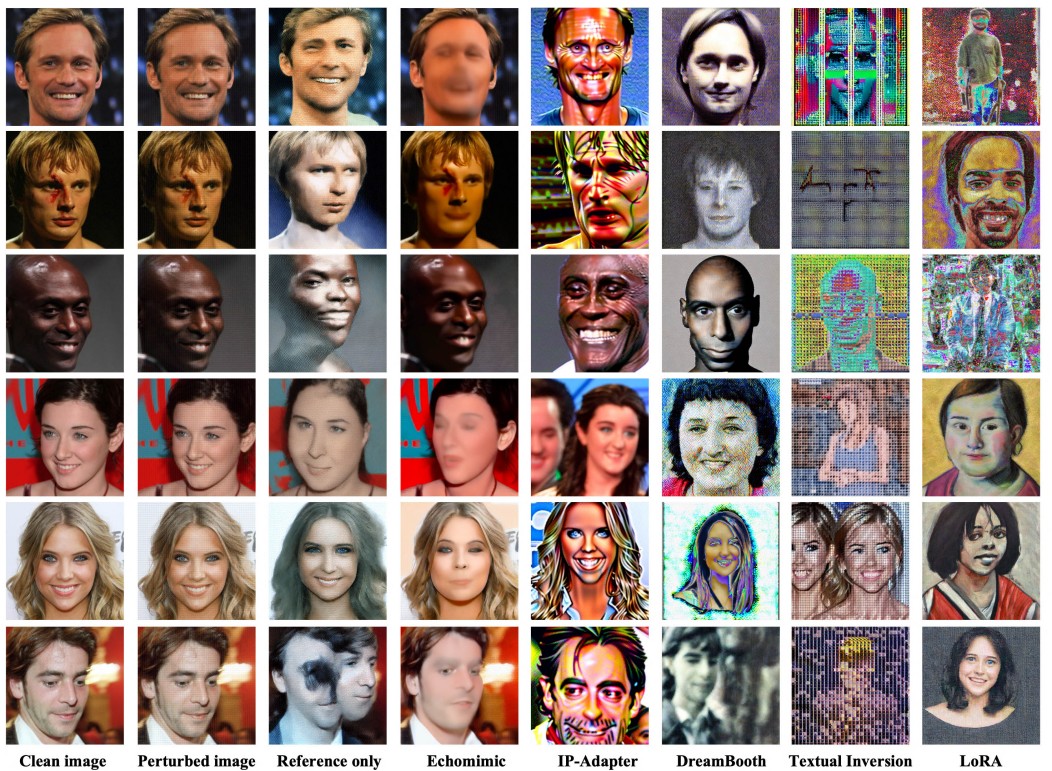

Figure 10: Additional results of our method (DiT).

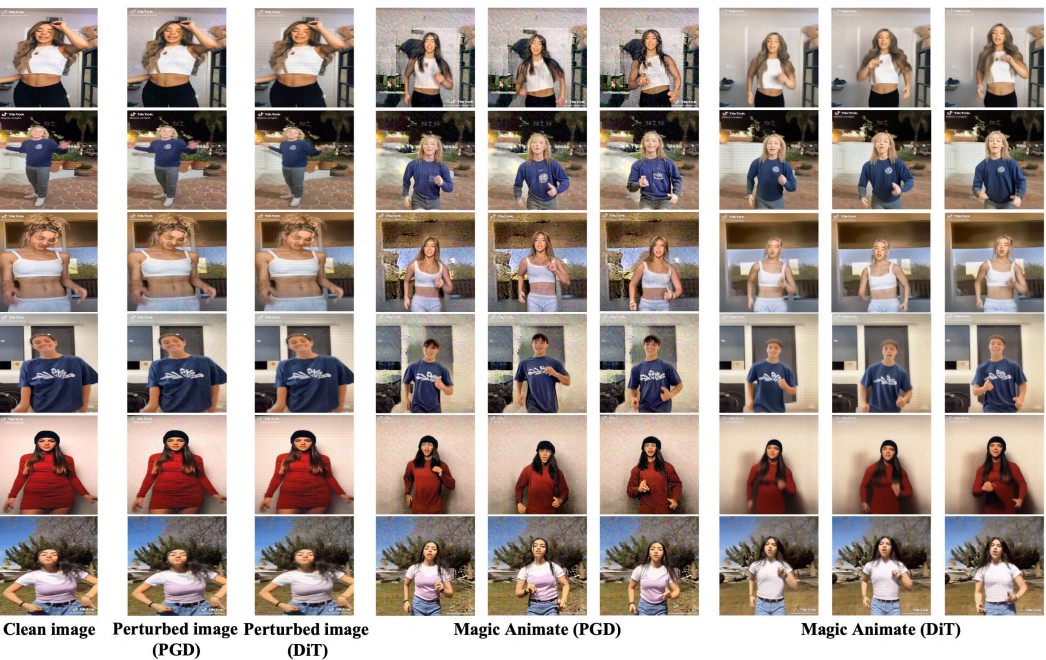

Figure 11: Additional results of our method, protect images against Magic Animate.

