# OpenReview forum: "Anti-Reference: Universal  and Immediate Defense Against Reference-Based Generation"
_ICLR.cc/2025/Conference — ICLR 2025 Conference Withdrawn Submission_

### Official Review · Reviewer_iRkg · 2024-10-31

**Soundness:** 4
**Presentation:** 3
**Contribution:** 2
**Rating:** 6
**Confidence:** 3

**Summary:**

This paper introduces an adversarial example-based defense approach against unauthorized image manipulations. Specifically, the authors propose a unified loss function that enables joint attacks on fine-tuning-based generation methods with a PGD-based adversary generation approach for optimization. Experiments across diverse datasets and settings are conducted.

**Strengths:**

1. The visual disruption effects are impressive, which demonstrates the effectiveness of privacy defence.
2. The proposed method is simple and effective.
3. The whole paper is generally well-written and organized.

**Weaknesses:**

1. This idea of using adversarial examples for privacy protection has been well-explored. Although this paper primarily focuses on defending privacy against diffusion-based models. Some related works for defending privacy against standard image generation models should also be discussed [a,b, c].
2. The possible defence mechanism built by the target generative models should be also considered, such as adversarial training and adversarial detection.

[a] Restricted black-box adversarial attack against deepfake face swapping
[b] Scalable Universal Adversarial Watermark Defending against Facial Forgery
[c] IDGuard: Robust, General, Identity-Centric POI Proactive Defense Against Face Editing Abuse

**Questions:**

1. Is it possible to provide more visualization samples for natural items instead of humans?
2. I'm curious about the gap between white-box and black-box attacks. How does the proposed method shrink this gap? An ablation study regarding it would be preferred

---

### Official Review · Reviewer_WZEH · 2024-11-03

**Soundness:** 2
**Presentation:** 2
**Contribution:** 2
**Rating:** 3
**Confidence:** 4

**Summary:**

This paper proposes an anti-reference method to protect images from imitation by reference-based generation techniques. It leverages a DiT-based Noise Encoder for efficient attacks without per-image optimization. The approach demonstrates effectiveness in defending against image imitation, with some evidence of transferability to black-box imitation methods.

**Strengths:**

1. The proposed framework’s use of a meta noise generator (the DiT noise encoder) to avoid optimization for each input image is logical and enhances efficiency. This efficiency gain has been validated in related research on attacking large language models [1].

[1] Paulus A, Zharmagambetov A, Guo C, et al. Advprompter: Fast adaptive adversarial prompting for llms[J]. arXiv preprint arXiv:2404.16873, 2024.

2. The method is straightforward and easy to implement.

**Weaknesses:**

1. Although the paper highlights the challenges in universality and transferability, it lacks an in-depth technical explanation of why existing methods fall short and how the proposed approach directly addresses these issues. The loss functions and optimization methods are not novel, which are standard in the prior works[2][3].

[2] Liang C, Wu X, Hua Y, et al. Adversarial example does good: Preventing painting imitation from diffusion models via adversarial examples[C]// ICML 2024.

[3] Van Le T, Phung H, Nguyen T H, et al. Anti-dreambooth: Protecting users from personalized text-to-image synthesis[C]//Proceedings of the IEEE/CVF International Conference on Computer Vision. 2023.

2. The experimental results are somewhat limited. The method’s primary application appears to be in the protection of copyrighted artwork; however, no dataset in this field is included.

**Questions:**

While the DiT-based noise generator is chosen, the rationale for its suitability remains unexplained. Are there specific optimization challenges with this architecture? Would joint optimization of the generator and reference network improve outcomes? With a fixed reference network and no adversarial training, there’s a risk that the noise generator may settle into trivial solutions, potentially compromising robustness.

---

### Official Review · Reviewer_6BU5 · 2024-11-04

**Soundness:** 2
**Presentation:** 2
**Contribution:** 2
**Rating:** 3
**Confidence:** 5

**Summary:**

This paper presents **Anti-Reference**, a defense mechanism designed to protect personal images from misuse by generative AI models. Anti-Reference applies imperceptible adversarial noise to images (in a way protected watermarking) to induce visible artifacts in images generated from them, thereby signaling these images as AI-generated and effectively protecting the originals from unauthorized manipulation.

Evaluation is done for two methods to optimize the adversarial noise;
a) Standard PGD Adversarial Attack is used to optimize the adversarial noise.
b) A novel DiT based Noise Encoder that is fine-tuned on a white-box model to apply adversarial noise.
The optimization of both PGD and Noise Encoder is done for three elements, defined in Total Loss on P7.

Qualitative results show that adding adversarial noise improves the protection of images against both white-box and *black-box* generative methods.

**Strengths:**

- The paper addresses a timely and significant issue: the protection of personal images from unauthorized use in reference-based AI generation. This aspect of privacy and ethical considerations in generative AI is critical and highly relevant in current research.

- Anti-Reference introduces a unique approach by combining adversarial noise with a DiT-based Noise Encoder, leveraging the transformer-based architecture's scalability for visual recognition tasks. This combination of techniques enhances the practicality and efficiency of image protection.

- In comparison to traditional PGD-based approaches, Anti-Reference significantly reduces computational time, thus increasing its feasibility for real-time applications. This efficiency improvement is well documented and enhances the method's practical value. Further, if fine-tuned correctly DiT based Noise Encoder can also be used to protect against *black-box* generative models.

**Weaknesses:**

- **Use of SDv1.5:** The Article only demonstrates the use case with the SDv1.5 Model, which is now deprecated, limiting the reproducibility of the results presented in the article. While most fine-tuned customizable models are based on the SDv1.5 model, it would be good if authors could demonstrate that the method can be applied to other Stable diffusion-based Models.

- **Usage of Black-Box Term:** The use of *black-box* is highly over-exaggerated. In a way, we know the architectures of the *black-box* methods used in the article, and the authors use the same architecture. In the field of adversarial machine learning, it is well-known that adversarial images transfer to similar architectures better. It would be good if the authors could show models protected with *SDv1.5* fine-tuning can still protect against *black-box* reference methods that are not based on *SDv1.5* (and even better if they are not Stable Diffusion).

- **Practical Applications:** The authors suggest that their method is real-world-ready (*albeit only for SDv1.5-based customization methods*), but there seems to be minimal analysis to support this argument. The only experiment that supports this argument is the use of JPEG Compression on the images (details are not provided regarding the amount of JPEG compression used). It is fair to say, in the real world, the images can be distorted before reaching the generative model; it can be resizing of the images, Strong JPEG Compression, Gaussian Blurring, Purifying the image by Diffusion Model by recreating the images (e.g., using Null-Text Inversion), or transformations applied by social-media platforms. Authors fail to verify whether adversarial noise is still effective in protecting the images after transformations that can happen in the real world. Current experiments are insufficient to argue that the method is practical in the real world.

- **Correcting the optimization problem:** Based on the optimization problem defined in Page 5, one has to increase the distance between Generated Images. However, this is not true for the losses they mention, as they do not increase the distance between the generated images. At the same time, it is intuitive that the distance between the generated images would increase for protected images since protected images would result in distorted images having a higher distance from the non-distorted images. Authors should bridge the gap between the optimization problem and their proposed loss theoretically; otherwise, one can argue that the protection is not optimization but a search.  Further, empirically, authors can also report the MSE between the generated images to demonstrate the distance between the generated images.

- **Reproducibility of Experiments:** The Authors introduce weights for each individual loss but fail to provide the values used in their experiments. Similarly, the article does not discuss the hyper-parameters of the PGD. To improve the reproducibility of the results, the authors should provide better experimental setup details.

- **Presentation of Qualitative Results:** Quantitative results are hard to interpret, and they show that existing methods and proposed methods perform similarly. First, it would be nice if Figure 4 (based on Tables 2, 3, 4, and 5 in the appendix) could be presented better. Second, there is no clear evidence that the methods perform better than the existing methods quantitatively. This creates suspicion about cherry-picking the qualitative results of whose analysis most of the article is based. Either the authors clearly explain in the main article why quantitative results contradict the qualitative results, or they demonstrate that their method is unarguably superior in quantitative metrics.

- **Limited Analysis:** Authors propose three individual losses but fail to analyze the impact of individual noise in protection. Further, there is no evidence to support why DiT-based noise encoders should be preferred over traditional ViT or CNNs. From the way the article is written, it feels like the authors stumbled upon a framework that works for protecting the images but did not analyze how it can be improved trivially (like changing architectures, modifying weights of individual losses, etc.). This makes the authors' proposal shallow and not insightful for the community.

**Questions:**

- It would be good if the authors numbered the Equations so that they can be referenced more clearly.

In Figure 2, $I_adv$ above the initial noise in the right part of the figure is confusing because it shows that $ I_adv$ is the Initial Noise. It would be better if the authors could move the term above the arrow to remove the confusion.

- Could the authors provide further technical details regarding the Noise Encoder's architecture, specifically the modifications made to adapt DiT for adversarial noise generation?

- What are the potential limitations of relying on the specific proxy models used in this study? Could the authors comment on the robustness of black-box transferability beyond the tested commercial APIs?


Overall, I feel the article has potential, but in its current form, it is not ready.

---

### Official Review · Reviewer_vAqm · 2024-11-04

**Soundness:** 3
**Presentation:** 3
**Contribution:** 3
**Rating:** 6
**Confidence:** 3

**Summary:**

The authors introduce Anti-Reference to protect images from reference-based generation by adding adversarial noise. They leverage a DiT-based Noise Encoder to generate the perturbation, with a joint loss term for enhancing the adversarial transferability across different types of customized models. The method maintains effectiveness against black-box models like commercial APIs.

**Strengths:**

- The paper is well-written and clearly presented with visual examples.
- The practical applicability of the proposed method is impressive, especially its success in some commercial APIs.

**Weaknesses:**

- **Applicability to New Models**: Anti-Reference is primarily tested on models like Stable Diffusion 1.5. The method may face challenges adapting to new architectures, such as Stable Diffusion XL or autoregressive models, as acknowledged in the limitations section.

**Questions:**

Have the authors tested Anti-Reference on any autoregressive generation models, and if so, how does it perform?

---

### Note · Authors · 2024-11-15

**Comment:**

Thank you for the valuable comments from the reviewers. We will revise the paper and resubmit it.

**Withdrawal Confirmation:**

I have read and agree with the venue's withdrawal policy on behalf of myself and my co-authors.